# Recurrence of WHO-defined fast breathing pneumonia among infants, its occurrence and predictors in Pakistan: a nested case–control analysis

Nick Brown [iD],[1,2] Arjumand Rizvi,[2] Salima Kerai,[3] Muhammad Imran Nisar,[2] Najeeb Rahman,[2] Benazir Baloch,[2] Fyezah Jehan [iD] [2]

¹International Maternal and Child Health, Department of Women's and Children's Health, Uppsala University, Academiska Sjukhuset, Uppsala, 75185, Sweden
²Department of Child Health, Aga Khan University Hospital, National Stadium Rd, Karachi, Sindh, 74800, Pakistan
³University of British Columbia, Vancouver, British Columbia, Canada

**Correspondence to**
Dr Nick Brown;
nick.brown@kbh.uu.se

## ABSTRACT

**Objectives** Studies in low-income and middle-income countries have shown an adverse association between environmental exposures including poverty. There is little literature from South Asia. We aimed to test the associations between housing, indoor air pollution and children's respiratory health and recurrent fast breathing pneumonia in a poor urban setting in Pakistan.

**Setting** Primary health centres in a periurban slum in Karachi, Pakistan.

**Methods** Nested matched case–control study within a non-inferiority randomised controlled trial of fast breathing pneumonia (Randomised Trial of Amoxicillin vs Placebo for Pneumonia (RETAPP)) in periurban slums of Karachi, Pakistan. Cases were children aged 2–60 months enrolled in RETAPP with fast breathing pneumonia who presented again with fast breathing between 8 weeks and 12 months after full recovery. Controls, selected in a 2:1 ratio, were age-matched participants who did not represent. Multivariable conditional logistic regression analysis was undertaken to explore associations with potentially modifiable environmental predictors including housing type, indoor air quality, exposure to tobacco smoke, outdoor pollution, household crowding, water and sanitation quality, nutritional status, immunisation completeness, breast feeding and airways hyperactivity.

**Results** Fast breathing recurred in 151 (3.7%) of children out of the total (4003) enrolled in the trial. Poor-quality housing of either katcha or mixed type strongly predicted recurrence with adjusted matched ORs 2.43 (95% CI 1.02 to 5.80) and 2.44 (1.11 to 5.38), respectively. Poor air quality, cooking fuel, inadequate ventilation, nutritional status, water, sanitation and hygiene (WASH) index, wheeze at first presentation and group of initial trial assignment were not independently predictive of recurrence.

**Conclusion** Poor-quality housing independently predicted recurrence of fast breathing pneumonia.

**Trial registration number** NCT02372461

## INTRODUCTION

Child pneumonia contributes approximately 30% of the total global pneumonia mortality[1–5] but, despite a gradual recent fall in both incidence and case fatality rates over the Millennium Development Goal era (1990–2015), it remains, by some distance, the single largest contributor to global post-neonatal mortality.[6 7]

A recent review for the Child Health Epidemiology Action Group estimated an incidence of community-acquired childhood pneumonia in low-income and middle-income countries (LMICs) of 0.22 (IQR 0.11–0.51) episodes per child per year of which 11.5% progress to severe episodes.[8] Recent data estimate 120 million illness episodes per year of which between 880 000 and 935 000 cases are fatal. Pneumonia, therefore, contributes of the order of 15% of the total 5.6 million annual under 5-year child deaths. The vast majority of the fatalities occur in LMICs in settings where resources are more stretched and populations more susceptible and most in the first 2 years of life.[6 9] It is a disease of poverty, strongly related to household overcrowding with undernutrition increasing case fatality.[10 11]

Childhood pneumonia is intimately linked to poverty in all its manifestations.[6] These

include overcrowding, undernutrition and the use of indoor fossil fuels.[10 11] The relative contribution of each is hard to ascertain epidemiologically as each of the factors is so closely linked to the others but have all been shown to be independent predictors.[6 7 12–17]

Though there is some literature on environmental predictors of recurrent pneumonia, there is little to date from Asia.[12 13] Given the prevalence of modifiable risk factors in periurban Pakistan, we aimed to establish the magnitude of their relative contributions in predicting recurrent episodes of WHO defined fast breathing pneumonia.[18]

## METHODS

We undertook a nested case–control study couched within the Randomised Trial of Amoxicillin vs Placebo for Pneumonia (RETAPP) study.[18 19] The methods and reporting were based on Strengthening the Reporting of Observational Studies in Epidemiology guidance (online supplementary file 1). In brief, RETAPP was a double blinded, randomised controlled non-inferiority trial undertaken between November 2014 and November 2017 in a poor periurban slum area (Bin Qasim town) in Karachi, Pakistan for which the Aga Khan University has provided demographic surveillance for many years. It tested the hypothesis that treatment failure rates in children aged 2–59 months with WHO-defined fast breathing pneumonia without chest indrawing[15] or other signs would be non-inferior in children given a placebo to those treated according to WHO guidance with a 3-day course of amoxicillin prescribed according to weight bands and approximating 45 mg/kg two times per day (WHO). A child met inclusion criteria if s/he presented with a respiratory rate of ≥50 breaths per minute (2–11 months) or ≥40 breaths per minute (12–59 months) with or without fever. A trained female health worker and a physician assessed respiratory rate independently and a child was designated a fast breather if there was agreement. Wheeze was assessed through auscultation by study a study physician. All children with wheeze received up to three doses of inhaled bronchodilator according to WHO-intefgrated manamgement of childhood illness (IMCI) guidelines.[18] Respiratory rate was reassessed after each inhalation. Only persistent fast breathers were retained. Children were excluded if there was associated lower chest indrawing, any danger signs and use of antibiotics in last 48 hours, hospitalisation in last 2 weeks, pedal oedema, known tuberculosis, asthma or other severe illness. The primary outcome was composite treatment failure 3 days after randomisation. A total of 4003 children were included in the primary analysis.

Infant mortality in the area is 78/1000 live births, one in six deaths attributable to pneumonia (demographic surveillance system (DSS), 2012). The major source of income in the area is fishing and manual labour. Average household income is low with 70% of the adult population earning less than US$5/day. Household size typically ranges from 5 to 11 members/household and about 12% of the dwellers use solid fuels in the form of open fire stoves, and wood burners or animal dung for indoor energy. The remainder use closed stoves and natural gas. Due to the long-standing presence of a demographic surveillance by the Aga Khan University through the primary healthcare centres, the majority of children with any illness are brought back by the caregivers for follow-up and management of their child's subsequent illnesses. The trial and case–control studies were based in four of the health centres.

### Nested case–control study

All previously randomised participants were eligible for inclusion in the nested case–control study A case was defined as a child previously randomised who represented with a recurrent episode of isolated fast breathing between 8 weeks and 1 year of the initial episode. Eight week was considered sufficient wash-out period for resolution of prior episode of fast breathing and the endpoint of a year was chosen for consistency with hospital studies of recurrent pneumonia.[19–21]

For case selection, children enrolled in RETAPP, who presented again with a recurrent fast breathing episode, were identified through their DSS identification number, date of birth and first name. Children with missing or incorrectly recorded identification numbers or non-matching date of birth of birth or first names between first and second visit were excluded. Controls were defined as children between 2 and 59 months of age, enrolled in RETAPP trial and who did not present with a recurrent episode of fast breathing during that defined time period. As the incidence of pneumonia is greater in younger children and because vaccination completion is age dependent, we matched each case with two controls of the same number of completed years of age. For each case, two matched controls were identified.

### Data collection

Exposure information for both cases and controls was obtained from their baseline visit (at the time of enrolment in RETAPP trial). Data were extracted for both cases and controls on several possible candidate predictors for fast breathing recurrence including: sex; sibling number; number of people reported to sleep in same room; exclusive breast feeding (up to 6 months of age); type of housing (katcha—made of mud, straw, wood and dry leaves, pakka-rooves, walls and floors made of high-quality material and mixed); WASH, status dichotomised as adequate or otherwise; nutrition status (based on WHO anthropometric z-scores of height for age and weight for age (a score of ≥ 2 SD being categorised as stunting and malnourished, respectively); completeness of vaccination (dichotomised, yes/no) defined according to the expanded programme for immunisation; the presence of wheeze at first presentation and the presence of pet animal in house. For completeness, we evaluated the association with the original group of randomisation to

assess any predictive or protective effect from earlier antibiotic treatment.

## Sample size and statistical analysis

The primary outcome was the recurrence of a fast breathing episode between 2 and 12 months after the index (RETAPP randomised) episode diagnosed, as in the main trial, by two independent assessments at presentation.

Sample size estimation was based on an OR of 2.5 by the presence of adverse environmental exposure representing a public health risk using an alpha of 5% and power of 80% (beta 20%). Based on pilot data of a 12% exposure prevalence of inadequate indoor air quality, in the population and a case–control ratio of 1:2, the minimum estimated sample size required was 106 cases and 211 controls, a total of 317 children. We ultimately included all 151 children with records of recurrence and 302 controls to enhance the precision of the estimated effect size.

All analyses were undertaken using STATA V.15 (StataCorp). Data were summarised using frequencies and proportions. The predictors of recurrent fast breathing were determined by multivariable conditional logistic regression after initial univariable analysis. Variables significant at p<0.2 in the univariable analysis were included in the fully adjusted model. The final model was constructed using backward elimination, variables being retained if p<0.1. Results were presented as adjusted matched ORs (aMORs). Plausible interaction and multicollinearity were additionally assessed. A p value <0.05 was deemed statistically significant.

To allow for any intraindividual non-independent correlations between observations in children with more than one recurrence presentation, we additionally undertook an exploratory sensitivity analysis of intraclass correlation coefficient (ICC). This showed a very low ICC (variance=0.0001, ICC=0) suggesting that each case could be treated as an independent variable. The final analysis, therefore, used simple conditional logistic regression rather than a mixed-effects model.

## Patient and public involvement

The concept of the original trial was introduced to parents in the communities served by the health centres before recruitment began, but, there was no specific patient or public involvement in the design of the study.

## RESULTS

Between November 2014 and November 2017, 53 838 children 2–59 months were self-referred to the primary healthcare centre. Of these, 7894 presented primarily with cough and tachypnoea and, of these, 4002 were eligible for and consented to enrolment in the RETAPP trial. Of these, 151 (3.7 %) met the criteria for eligibility in terms of a recurrence between 8 weeks and 1 year of initial enrolment (cases) (figure 1). There was no clear seasonal

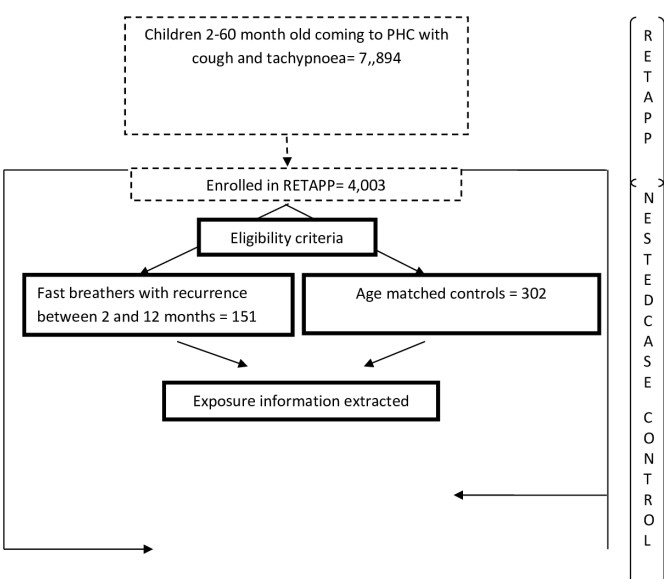

**Figure 1** Study flow chart. RETAPP, Randomised Trial of Amoxicillin versus Placebo for Pneumonia; PHC, primary health centre.

pattern to incidence of recurrent pneumonia (online supplementary file 2). We selected 302 children who did not experience a recurrent episode (a 2:1 ratio) matched by age band. The selected controls were compared with the non-selected children without recurrence and shown to be similar in terms of baseline variables (online supplementary file 2).

Univariable analysis (table 1) showed a significant association with poorer housing type and drinking water and protective effect of the presence of a household smoker and incomplete vaccination. In the multivariable analysis (table 2), only poor-housing quality remained predictive: with aMORs (95% CI) relative to the pukka housing reference of: katcha 2.43 (1.02 to 5.80, p=0.04) and mixed 2.44 (1.11 to 5.38, p=0.03), respectively. There was no association with: undernutrition; sex; inadequate ventilation; absence of breast feeding; type of household fuel used; sibling number; bedroom crowding; wheeze at first presentation; household pet animals; sanitation or the proximity of the child's school to factories, traffic fumes or public ovens.

## DISCUSSION

Our findings, to our knowledge, are the first to show the burden of recurrence of fast breathing pneumonia and the association with poor-housing quality in this setting. A total of 8.3% of the children lived in 'katcha' or mixed housing, forms of mud brick buildings often made as temporary homes for the poorest families which, even by urban slum standards, are rudimentary

In the multivariable analysis, poor-housing quality was independently associated with recurrence with aMORs (95% CI) relative to the pukka housing reference of 2.43 (1.02 to 5.80, p 0.04) and mixed 2.44 (1.11 to 5.38, p

**Table 1** Baseline demographic variables in cases and controls

|  | Control | Case |
|---|---|---|
|  | n=302, (%) | n=151, (%) |
| **Child sex** |  |  |
| Male | 167 (55.3) | 83 (55.0) |
| Female | 135 (44.7) | 68 (45.0) |
| **Sibling no** |  |  |
| ≤3 | 218 (72.2) | 108 (71.5) |
| >3 | 84 (27.8) | 43 (28.5) |
| **Room sharing no** |  |  |
| ≤4 | 173 (57.3) | 81 (53.6) |
| >4 | 129 (42.7) | 70 (46.4) |
| **Housing type** |  |  |
| Katcha | 10 (3.3) | 13 (8.6) |
| Mixed | 15 (5.0) | 16 (10.6) |
| Pakka | 277 (91.7) | 122 (80.8) |
| **Used improved drinking water source** | 274 (90.7) | 128 (84.8) |
| **Used improved sanitation facilities** | 274 (90.7) | 135 (89.4) |
| **Cooking fuel** |  |  |
| Gas | 261 (86.4) | 126 (83.4) |
| Wood/coal/animal dung | 41 (13.6) | 25 (16.6) |
| **Location of child's school** |  |  |
| Otherwise | 153 (56.0) | 70 (49.6) |
| Near factory/traffic fumes/public oven | 120 (44.0) | 71 (50.4) |
| N | 273 | 141 |
| **Proper ventilation in house** | 274 (90.7) | 140 (92.7) |
| **Current smoker in household** | 70 (23.2) | 22 (14.6) |
| **Pets in household** | 74 (24.5) | 32 (21.2) |
| **Wheeze at presentation** | 23 (7.6) | 12 (7.9) |
| **Underweight** | 110 (36.4) | 60 (39.7) |
| **Stunting** | 120 (40.1) | 71 (47.3) |
| N | 299 | 150 |
| **Wasting** | 51 (17.0) | 29 (19.2) |
| N | 300 | 151 |
| **Received age adequate vaccination** | 134 (44.4) | 84 (55.6) |
| **History of exclusive breast feeding** | 187 (61.9) | 93 (61.6) |

Katcha housing: made up of mud, straw, wood and dry leaves.
Pakka housing: rooves, walls and floors made of high-quality material.
Mixed: a combination of pakka and katcha.

0.03) for katcha and mixed housing, respectively. There was no association with: undernutrition; sex; inadequate ventilation; absence of breast feeding; type of household fuel used; sibling number; bedroom crowding; wheeze at first presentation; household pet animals; sanitation or the proximity of the child's school to factories, traffic fumes or public ovens.

We feel the study has a number of strengths. First, though there is some published work from LMICs related to environment and respiratory health, there has been little data from Asia.

Second, as a result of the existing infrastructure in our population, follow-up was robust and capture of recurrence of pneumonia was excellent.

Third, the matched design removes potential confounding effects of age and immunisation status. Infants have higher incidence rates of pneumonia than older children and, for logistical reasons are more likely not to have completed routine vaccination. Matching is likely to have improved the precision of the estimate.

The study and inferences have some limitations. First, the population is relatively homogeneous in terms of socioeconomic status, which may have blunted estimated effect sizes of some potential exposures. For example, all households were poor in comparison to middle class inner city dwellers and some degree of indoor air pollution exposure was the norm. Crude nutritional status by weight for age z-score was also prevalent (underweight 36.4% and stunted 40%) that any differences that were might have been falsely negative. Second, there was lack of database linkage due to incorrectly recorded demographic surveillance identity in a small number (17.7%) of cases. Third, it is also possible that a small proportion of recurrent cases may have presented to other facilities (private or out of the city), which may have led to an underestimate of incidence of recurrence. Fourth, we did not directly measure particulate matter density to quantify solid fuel exposure but used a questionnaire which might have been too insensitive to detect differences. Finally, we have inferred that housing quality is a valid proxy for poverty. While we feel this is appropriate, it might be a marker of other unmeasured exposures such as migration or recent displacement from permanent homes.

Childhood pneumonia is intimately linked to poverty in all its manifestations. These include overcrowding, undernutrition, incomplete vaccination and the use of indoor air pollution.[11 12 17] Sonego et al's meta-analysis estimated effect sizes for a range of socioeconomic predictors and found that early pneumonia mortality was significantly related to young maternal age, low maternal education and low socioeconomic status.[12]

Poverty is a sensitive, marker for adverse environmental stress.[1–8 10–14 16] Not all poverty phenotypes are as easily measurable as, for example, the 'classic' markers: nutritional status; household size; completeness of vaccination; exposure to contaminated drinking water and airborne fine particulate matter. There is substantial evidence that one pathway is of chronic low-grade inflammation demonstrable, for example, by raised mean C reactive protein in more deprived populations[22] and of subtle micronutrient deficiencies and low level intergenerational psychological stress[23] all of which can be immune suppressant. Other factors increasing susceptibility include the mediating effects of low maternal education, maternal depression and impaired mother child interaction on childhood illness.[24]

**Table 2** Matched analysis showing effect of predictors on recurrent fast breathing

| Predictors | Unadjusted | | Adjusted | |
|---|---|---|---|---|
| | Matched OR (95% CI) | P value | Matched OR (95% CI) | P value |
| **Randomisation group** | | | | |
| Amoxicillin | 1.29 (0.87 to 1.91) | 0.205 | – | – |
| Placebo | Ref. | | Ref. | |
| **Child sex** | | | | |
| Male | Ref. | | Ref. | |
| Female | 1.01 (0.69 to 1.50) | 0.947 | – | – |
| **Sibling no** | | | | |
| ≤3 | Ref. | | Ref. | |
| >3 | 1.03 (0.67 to 1.59) | 0.883 | – | – |
| **Room sharing no** | | | | |
| ≤4 | Ref. | | Ref. | |
| >4 | 1.15 (0.78 to 1.70) | 0.469 | – | – |
| **Housing type** | | | | |
| Katcha | 2.80 (1.22 to 6.44) | 0.015 | 2.43 (1.02 to 5.80) | 0.045 |
| Mixed | 2.53 (1.18 to 5.46) | 0.018 | 2.44 (1.11 to 5.38) | 0.026 |
| Pakka | Ref. | | Ref. | |
| **Unimproved drinking water** | 1.86 (0.99 to 3.49) | 0.053 | 1.72 (0.89 to 3.36) | 0.109 |
| **Unimproved sanitation facilities** | 1.16 (0.61 to 2.21) | 0.655 | – | – |
| **Cooking fuel** | | | | |
| Gas | Ref. | | Ref. | |
| Wood/coal/animal dung | 1.28 (0.73 to 2.22) | 0.387 | – | – |
| **Location of child school** | | | | |
| Otherwise | Ref. | | | |
| Near factory/traffic fumes/public oven | 1.28 (0.84 to 1.95) | 0.253 | – | – |
| **No proper ventilation in house** | 0.78 (0.38 to 1.59) | 0.487 | – | – |
| **Current smoker in household** | 0.57 (0.34 to 0.96) | 0.033 | 0.53 (0.30 to 0.91) | 0.022 |
| **Pets in household** | 0.83 (0.52 to 1.33) | 0.433 | – | – |
| **Wheeze at presentation** | 1.05 (0.50 to 2.18) | 0.901 | – | – |
| **Underweight** | 1.16 (0.77 to 1.74) | 0.486 | – | – |
| **Stunted** | 1.39 (0.91 to 2.14) | 0.129 | – | – |
| **Wasted** | 1.16 (0.70 to 1.94) | 0.566 | – | – |
| **Age adequate vaccination** | | | | |
| No | 0.62 (0.41 to 0.93) | 0.021 | 0.61 (0.40 to 0.94) | 0.025 |
| Yes | Ref. | | Ref. | |
| **History of exclusive breast feeding** | | | | |
| Yes | Ref. | | Ref. | |
| No | 1.01 (0.68 to 1.52) | 0.946 | – | – |

Recent Global Burden of Disease analysis indicated that poverty was the single largest contributor to childhood illness in sub-Saharan Africa[25] though any association between poverty, education, family size, sanitation and fuel use is complex. In recent work in Pakistan, Nasir *et al* showed multiple inter-related pathways.[26] In Ram *et al*'s study in Bangladesh in children aged under 5 years with pneumonia where comparable proportions to our study lived in households using gas (85% vs 86% in Karachi) suggested that improved ventilation and the reduction of overcrowding were the single largest preventable determinants of illness. Tin roofs, low socioeconomic status and male sex were other predictors, but, as in our study particulate matter per se was not.[27]

The relative contribution of each airborne pollutant is arguably harder to ascertain epidemiologically as each

of the factors, black carbon, fossil fuel, fine particulate matter (PM 2.5) and carbon monoxide (CO), is so closely linked to all the others[7] Extrapolating this, Adaji's meta-analysis showed no association between home CO exposure and pneumonia. Fine particulate matter was only significant when using fuel as a proxy for exposure and not when measured directly highlighting the issues with standardising exposure measurement in such settings.[16]

It would be consistent with this hypothesis that household size is too blunt a tool, the majority of the families consisting of 10 or more members and multiple room sharing inevitable. Incomplete vaccination status similarly was so common (55.6%) that interpretation of the result difficult and likely to be explained by confounding. In addition, most episodes of fast breathing pneumonia are known to be viral against which the common capsulated polysaccharide antigens are unlikely to be protective. Given the association of poverty with recurrent pneumonia and the cost of cigarettes, it seems probable that the effect of household smoking is confounded by wealth. Consistent with this assertion is the association between household smoking and household wealth.

There are potential policy implications of these findings. Pneumonia is a common childhood illness and higher incidence rates likely to be a marker of general poor health. Though the associations between environmental exposures and poor WASH status are well recognised, we found an independent association with poor housing. Katcha and mixed housing are inextricably bound to poverty and might also mirror subtle nutritional deficiencies and family displacement. It is possible that provision of higher quality housing might ameliorate this effect. There is also a case for enhanced routine health surveillance in these families. This would not be without complexity as poorer families are less likely to present to health facilities and door-to-door monitoring might be required.

## CONCLUSION

Recurrent fast breathing pneumonia in this deprived periurban population in Pakistan was significantly independently associated with poor-housing quality. It is likely that rudimentary homes are a marker of poverty. Factors other than recognised environmental exposures but inherent to poverty are likely to be involved.

**Acknowledgements** We would like to acknowledge patients and families, the RETAPP study team, including community health workers, demographic surveillance workers and the committed physicians who took care of patients.

**Contributors** FJ conceived the idea; FJ, SK, MIN, BB and NB designed the study. FJ, MIN, BB, NB and NR were involved in conduct of the study. NB, AR, FJ, SK, MIN, NB and NR were involved in statistical analysis. FJ, NB and SK wrote the first draft of the manuscript and NB rewrote new drafts based on input from coauthors MIN, NB, BB and NR. All authors read and approved the final manuscript.

**Funding** This work is jointly funded by the MRC-Wellcome-DFID through the Joint Global Health Trials, (grant number MR/L004283/1) and the Bill & Melinda Gates Foundation (grant number OPP1158281). FJ and MIN received funding from the National Institute of Health's Fogarty International Center (grant number 1 D43 TW007585-01).

**Competing interests** None declared.

**Patient consent for publication** Not required.

**Ethics approval** Ethical approval for the original trial was obtained both from the Aga Khan University (2786-pee-ERC-13). As the original trial was initially UK funded, we also obtained endorsement from the University of Southampton ethics' committee (ID 8700- December 2013).

**Provenance and peer review** Not commissioned; externally peer reviewed.

**Data availability statement** Data are available on reasonable request.

**Open access** This is an open access article distributed in accordance with the Creative Commons Attribution 4.0 Unported (CC BY 4.0) license, which permits others to copy, redistribute, remix, transform and build upon this work for any purpose, provided the original work is properly cited, a link to the licence is given, and indication of whether changes were made. See: https://creativecommons.org/licenses/by/4.0/.

**ORCID iDs**
Nick Brown http://orcid.org/0000-0003-1789-0436
Fyezah Jehan http://orcid.org/0000-0002-5874-4358

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
