## [Reviewer comments · BMJ Open]

ARTICLE DETAILS

TITLE (PROVISIONAL)	Recurrence of World Health Organization defined Fast Breathing Pneumonia among Infants, its Occurrence and Predictors in Pakistan- A Nested Case Control Analysis.
AUTHORS	Brown, Nick; Rizvi, Arjumand; kerai, salima; Nisar, Muhammad; Rahman, Najeeb; Baloch, Benazir; Jehan, Fyezah

VERSION 1 – REVIEW

REVIEWER	Dr Megha Thakur Maastricht University CAPHRI School for Public Health and Primary Care
REVIEW RETURNED	21-Aug-2017

GENERAL COMMENTS	The manuscript was interesting to read, and is concise and organized. Below are some comments from my side- MAIN TEXT • Background: I think the introduction is fundamentally sound, updated, reflecting the importance of the recurrent pneumonia and the paucity of evidence, especially from developing countries.• Methods: Just curious to know if the case definition was supported by radiological findings, or it was just based on recurrent episode of isolated fast breathing between eight weeks to one year of the initial episode. Was any assessment made based on the season of the year? Were there more cases of RP in winters or rainy season? Was any information collected on underlying conditions such as, family history of asthma, congenital cardiac defects, immunological abnormalities, premature delivery, and neuromuscular disorders etc.? Was temporary migration in the slum taken into account? Could the authors explain what is meant by an open and closed stove? Does closed stove refer to stove using natural gas, or it is a conventional stove, but not an open fire? In Table 1 (page 11), the proportion of cases and controls using an open stove is quite high, 94% and 89% respectively, but the solid fuel use is only 24% and 12% respectively. I don't understand the mismatch. Could this be clarified? As the parent trial has been administering oral amoxicillin to the children with fast breathing, were there any children who didn't complete the 3 day course? Were these children part of the nested case-control analysis? Sample size calculation is not clear enough. Why was solid fuel use chosen as the main predictor for RP? Was there any specific hypotheses?
--

	Overall, the methods section needs clarification. • Results: The results mention that only solid fuel use independently predicted recurrent fast breathing, however, in the absence (in case) of assessment of other potential predictors as mentioned above that could have important implications, the results must be interpreted carefully. The authors have transparently reported the limitations of the study, including small sample size, few cases due to lack of database linkage, underestimation of recurrent cases due to presentation to other health facilities, and inability to quantify solid fuel exposure.
--	--

REVIEWER	Thi Kim Phuong Nguyen Da Nang hospital for Women and Children, Viet Nam
REVIEW RETURNED	17-Sep-2017

GENERAL COMMENTS	This is an interesting study, which provided the audience some risk factors of recurrent pneumonia. Some risk factors of child pneumonia are recognized by WHO but unsure their role in recurrent pneumonia. The study showed a strong correlation between indoor air pollution exposure with increase the risk and completed vaccination as a protected factor. It will be more interesting if the authors considered 2 other important risk factors as exclusive breastfeeding and co-morbid conditions in these recurrent cases. Recurrence of World Health Organization defined Fast Breathing Pneumonia among Infants, its Occurrence and Predictors in Pakistan- A Nested Case Control Analysis. Comments This is an interesting study, which provided the audience with suggested predictors of recurrent pneumonia. Some risk factors of child pneumonia are recognized by WHO but unsure their role in recurrent pneumonia. The study showed a strong correlation between indoor air pollution exposure with increase the risk and completed vaccination as a protected factor. It will be more interesting if the authors considered 2 other important risk factors as exclusive breastfeeding and co-morbid conditions in these recurrent cases. I just have minor suggestions Introduction section Page 5: It that possible if the authors just use pneumonia or ALRI instead of them both in the text? This may sometimes be confused. Line 22, page 5: in revised WHO pneumonia classification 2014, “non-severe pneumonia” = fast breathing +/- chest indrawing; “severe pneumonia” = fast breathing + any danger signs. Please check your text to make this clear and consistent. Method section Some points that may affect your results and need to be considered  1. Did these cases visit and receive any treatment from private doctor, pharmacist, traditional healer before admitting to the PHC? 2. Did they go to child care or had any siblings? If that, was there any sick children (with ARI symptoms) at school or home? 3. History of TB contact, etc... Result section
---

	I would prefer a confidence interval belongs to each OR in table 1 Discussion section Consider these points in method section that I suggested above Please delete 1 of 2 “from” in line 11, page 15 Reference You can find the update number of child pneumonia disease burden from this article Li Liu, Shefali Oza, Dan Hogan, Yue Chu, Jamie Perin, Jun Zhu, et al. Global, regional, and national causes of under-5 mortality in 2000–15: an updated systematic analysis with implications for the Sustainable Development Goals. Lancet. 2016;16:31593-8.
--	--

REVIEWER	Carina King University College London, UK
REVIEW RETURNED	18-Sep-2017

GENERAL COMMENTS	This is an important topic, and one which definitely needs more investigation, so this is a relevant paper. It is well presented and clear, and I only have minor comments for clarification and two requests for a bit more analysis and data to be presented. A rigorous proof-reading is also needed. Strengths & Limitations: - 'A priori' Introduction: - In the description of ALRI you discuss chest in-drawing pneumonia as severe. This needs to be made a bit clearer, as the revised IMCI guidelines have chest in-drawing pneumonia being treated as an outpatient (pneumonia rather than severe pneumonia). Needs a clarification about which guidelines you are using. Methods: - Study design and setting: Was chest in-drawing included in the danger signs? It might be useful to list the danger signs just to clarify for the reader (e.g. wheeze?). - Study design and setting: Who conducted the follow-ups, were they home or facility based? - Study population: it would be useful to know a bit more about where geographically the study was conducted. - Data collection: how was vaccine status recorded, from documented records, or caregiver recall? And was wheeze assessed by the study physician, or was this care-giver reported history? Results: - Were the randomly selected controls representative of all the potential controls? A supplementary table would be useful. - For the multivariate analysis, what variables were included in the model? - Was age a significant predictor in univariate analysis? And was this collinear with vaccination completion? It would be interesting to see if that association was mediated by age at all. Discussion: - 'Our findings are the foremost to show', change to 'Our findings are the first to show' - Which formulation of PCV was introduced? PCV13?
---

	- Re-phrase 'wheeze at presentation was protective' to 'wheeze at presentation was less frequent' Table: - To make it easier to read the table, all the reference categories should be at the top for each variable.
--	---

VERSION 1 – AUTHOR RESPONSE

Reviewers' Comments to Author:

Reviewer: 1

Reviewer Name: Dr Megha Thakur

Institution and Country: Department of Family Medicine, Care and Public Health Research Institute (CAPHRI), Maastricht University, the Netherlands

Competing Interests: None declared

1. The manuscript was interesting to read, and is concise and organized.

Response

Thank you

Below are some comments from my side-

Main text

2. Background:

I think the introduction is fundamentally sound, updated, reflecting the importance of the recurrent pneumonia and the paucity of evidence, especially from developing countries.

Response

Thank you

3. Methods

Just curious to know if the case definition was supported by radiological findings, or it was just based on recurrent episode of isolated fast breathing between eight weeks to one year of the initial episode.

Response

In keeping with the WHO pragmatic syndromal management of fast breathing pneumonia, no x rays were undertaken in children without complications

4. Was any assessment made based on the season of the year? Were there more cases of RP in winters or rainy season?

Response

Thank you. Supplementary table 1 shows the seasonal distribution both in tabular and graphic form. There was no clear seasonal pattern

5. Was any information collected on underlying conditions such as, family history of asthma, congenital cardiac defects, immunological abnormalities, premature delivery, and neuromuscular disorders etc.?

Response

Exclusion criteria were:

- Antibiotic treatment within the last 48 h
-

- WHO defined danger sign (stridor when calm, hypoxia defined as $\text{SaO}_2 < 90\%$ in air, inability to feed, persistent vomiting, convulsions or reduced conscious level, bulging fontanel, pedal edema
-
- Known asthma or the normalization of fast breathing after the administration of the bronchodilator salbutamol
- Tuberculosis or other severe illness: this would include known neuromuscular and immunological defects
-
- History of hospitalization in last two week
-
- Known congenital heart disease
-
- Any surgical condition requiring hospitalization
-
- Out of catchment area
-
- Enrolled in another trial or previously enrolled in study

6. Was temporary migration in the slum taken into account?

Response

Migration was not an issue as to warrant inclusion, the families of all children would have been resident for at least 6 months prior to the initial presentation

7. Could the authors explain what is meant by an open and closed stove? Does closed stove refer to stove using natural gas, or it is a conventional stove, but not an open fire? In Table 1 (page 11), the proportion of cases and controls using an open stove is quite high, 94% and 89% respectively, but the solid fuel use is only 24% and 12% respectively. I don't understand the mismatch. Could this be clarified?

Response

We have simplified our predictors to include only solid (wood, coal and animal dung) and gas

8. As the parent trial has been administering oral amoxicillin to the children with fast breathing, were there any children who didn't complete the 3 day course? Were these children part of the nested case-control analysis?

Response

No. Only children who completed treatment in the trial were eligible

9. Sample size calculation is not clear enough. Why was solid fuel use chosen as the main predictor for RP? Was there any specific hypotheses?

Overall, the methods section needs clarification.

Response

Thanks for this helpful comment. The study was in part exploratory and hypothesis generating, but, to inform sample size we made an estimate based on a posited effect size (OR 2.5) of public health importance for the composite indoor air pollution variable on recurrent pneumonia

10 Results

The results mention that only solid fuel use independently predicted recurrent fast breathing, however, in the absence (in case) of assessment of other potential predictors as mentioned above that could have important implications, the results must be interpreted carefully.

The authors have transparently reported the limitations of the study, including small sample size, few

cases due to lack of database linkage, underestimation of recurrent cases due to presentation to other health facilities, and inability to quantify solid fuel exposure.

Response

Thank you. We have been careful to avoid over interpretation of the results

Reviewer: 2

Reviewer Name: Thi Kim Phuong Nguyen

Institution and Country: Da Nang hospital for Women and Children, Viet Nam

Competing Interests: None to declared

See file attached.

1. This is an interesting study, which provided the audience some risk factors of recurrent pneumonia. Some risk factors of child pneumonia are recognized by WHO but unsure their role in recurrent pneumonia. The study showed a strong correlation between indoor air pollution exposure with increase the risk and completed vaccination as a protected factor. It will be more interesting if the authors considered 2 other important risk factors as exclusive breastfeeding and co-morbid conditions in these recurrent cases.

Response

Thank you. Breast feeding was not predictive or protective in the univariable model so was not included in the multivariable analysis. Breast feeding is however, the norm so differences might be harder to detect

...

2. It that possible if the authors just use pneumonia or ALRI instead of them both in the text? This may sometimes be confused.

Response

Thank you. We have now used the term pneumonia throughout

3. In the revised WHO pneumonia classification 2014, “nonsevere pneumonia” = fast breathing +/- chest indrawing; “severe pneumonia” = fast breathing + any danger signs. Please check your text to make this clear and consistent.

Response

Thank you. We have reworded this section

4. Did these cases visit and receive any treatment from private doctor, pharmacist, traditional healer before admitting to the PHC?

Response

We specifically enquired about additional treatment. Any children that had received antibiotics for any indication within 48 hours of presentation were excluded

5. Did they go to child care. Was sibling number addressed. Were there sick children at home. History of TB contact

Response

It is traditional for all children in this community to be cared for at home

We do not have data on concurrent illness in siblings

Sibling number was included as a co-variate in the regression model. It was not significant

Known tuberculosis was an exclusion criterium

6. I would prefer a confidence interval belongs to each OR in table 1

Response

These have now been included

7. Delete 'from' line 11, page 15

Response

Deleted

8. Additional reference for child pneumonia disease burden from
this article

Response

Thank you. We have included the Liu, Lancet 2016 reference as suggested

Li Liu, Shefali Oza, Dan Hogan, Yue Chu, Jamie Perin, Jun Zhu, et al.

Global, regional, and national causes of under-5 mortality in 2000–15: an updated systematic analysis with implications for the Sustainable Development Goals. *Lancet*. 2016;16:31593-8.

Reviewer: 3

Reviewer Name: Carina King

Institution and Country: University College London, UK

Competing Interests: None declared

1. General

This is an important topic, and one which definitely needs more investigation, so this is a relevant paper. It is well presented and clear, and I only have minor comments for clarification and two requests for a bit more analysis and data to be presented. A rigorous proof-reading is also needed

Response

Thank you

Strengths & Limitations:

- 'A priori' -corrected

2. Introduction

In the description of ALRI you discuss chest in-drawing pneumonia as severe. This needs to be made a bit clearer, as the revised IMCI guidelines have chest in-drawing pneumonia being treated as an outpatient (pneumonia rather than severe pneumonia). Needs a clarification about which guidelines you are using.

Response

Thank you. Clarified

3. Methods

- Study design and setting: Was chest in-drawing included in the danger signs? It might be useful to list the danger signs just to clarify for the reader (e.g. wheeze?).

Response

Thank you. We have included more information about the setting

In terms of methodology we have added

' A child met inclusion criteria if s/he presented with a respiratory rate of ≥ 50 breaths per minute (2-11 months) or ≥ 40 breaths per minute (12-59 months). A trained lady health worker and a physician assessed respiratory rate independently and a child was labeled as fast-breather if there was agreement. Wheeze was assessed through auscultation by study physician, all children with wheeze received up to three doses of inhaled bronchodilator according to WHO-IMCI guidelines. Respiratory rate was recounted and child was re-categorized for fast breathing after each inhalation. Only persistent fast breathers were retained. Children were excluded if there was associated lower chest indrawing, any danger signs, and use of antibiotics in last 48 hours, hospitalization in last two weeks, pedal edema, known tuberculosis, asthma or other severe illness'

- Study design and setting: Who conducted the follow-ups, were they home or facility based?

Response

Thank you

In the RETAPP trial follow ups which included directly observed treatment and a clinical assessment were conducted by physicians in the mornings and community health workers in the evenings

- Study population: it would be useful to know a bit more about where geographically the study was conducted.

Response

Detail in methods

- Data collection: how was vaccine status recorded, from documented records, or caregiver recall? And was wheeze assessed by the study physician, or was this care-giver reported history?

Response

Thank you

Vaccination status was recorded from hand held records and (where unavailable) parental recall

Wheeze was assessed by study physicians. If present, a child was given three doses of a bronchodilator in accordance with WHO guidance. If the respiratory rate normalized, the child was deemed ineligible

Response

4. Results

- Were the randomly selected controls representative of all the potential controls? A supplementary table would be useful.

Response

Thank you. Supplementary table 1 shows that the selected controls were comparable to all non-recurrent pneumonia children

- For the multivariate analysis, what variables were included in the model?

- Was age a significant predictor in univariate analysis? And was this collinear with vaccination completion? It would be interesting to see if that association was mediated by age at all.

Response

Thank you

The matched controls were representative- see supplementary table

See table 2: housing type, poor water, household smoking and incomplete vaccination

We have completely reworked the analysis with new controls and matched for age on the basis of likely collinearity with vaccination status and the known increased incidence of pneumonia in infancy. Age does, therefore, not appear

5. Discussion

- 'Our findings are the foremost to show', change to 'Our findings are the first to show'
- Which formulation of PCV was introduced? PCV13?
- Re-phrase 'wheeze at presentation was protective' to 'wheeze at presentation was less frequent'

Response

Thank you. Rewritten on the basis of the new matched analysis

6. Table:

- To make it easier to read the table, all the reference categories should be at the top for each variable

Response

Thank you. Amended

VERSION 2 – REVIEW

REVIEWER	Thi Kim Phuong Nguyen Da Nang Hospital for Women and Children, Vietnam
REVIEW RETURNED	31-Oct-2019

GENERAL COMMENTS	Thanks for addressing most of my previous comments. I have a few suggestions as below: 1. Abstract Line 15: I am not sure if "poverty" could be part of "environmental exposures". Write abbreviations out for the first time: MOR, WASH 2. Introduction Write abbreviations out for the first time: LMIC, WHO... 3. Methods Would you exclude children with fever?
--

	4. Results Please adding footnote under each tables. Definition of Katcha, Pakka
--	--

REVIEWER	Carina King University College London
REVIEW RETURNED	11-Nov-2019

GENERAL COMMENTS	Thank you for addressing my previous comments - I think the methods are now clearer, and using a matched analysis has improved the results. There are still a few minor grammatical errors, and I think one of the < or > are the wrong way around. In the results, poor housing is described as katcha and mixed - but these need to be explained. It would be better to express as "associations with" rather than "predictive of" in the results.
--

VERSION 2 – AUTHOR RESPONSE

Reviewers' Comments to Author:

Reviewer: 1

Reviewer Name: Thi Kim Phuong Nguyen

Institution and Country: Da Nang Hospital for Women and Children, Vietnam

Please state any competing interests or state 'None declared': None declared

Thanks for addressing most of my previous comments. I have a few suggestions as below:

1. Abstract

Line 15: I am not sure if "poverty" could be part of "environmental exposures".

Response

Thank you. We used the term in a broad sense which we felt was justified. However, we can understand the alternative interpretations and have therefore removed it

Write abbreviations out for the first time: MOR, WASH

2. Introduction

Write abbreviations out for the first time: LMIC, WHO..

Response

Thank you. Amended

3. Methods

Would you exclude children with fever?

Response

Thank you. All children fulfilling the WHO fast breathing criteria irrespective of whether or not they had fever at presentation were eligible. We have clarified this in the methods

4. Results

Please adding footnote under each tables.

Definition of Katcha, Pakka

Response

Thank you. A very helpful suggestion

Reviewer: 2

Reviewer Name: Carina King

Institution and Country: University College London, UK

Please state any competing interests or state 'None declared': None declared

Thank you for addressing my previous comments - I think the methods are now clearer, and using a matched analysis has improved the results. There are still a few minor grammatical errors, and I think one of the < or > are the wrong way around. In the results, poor housing is described as katcha and mixed - but these need to be explained. It would be better to express as "associations with" rather than "predictive of" in the results.

Response

Thank you for the generous comments

Amended as suggested